# Mamwi Gidaanjitoomin/Together We Build It: A Systematic Review of Traditional Indigenous Building Structures in North America and Their Potential Application in Contemporary Designs to Promote Environment and Well-Being

**DOI:** 10.3390/ijerph20064761

**Published:** 2023-03-08

**Authors:** Angela Mashford-Pringle, Ruofan Fu, Sterling Stutz

**Affiliations:** 1Waakebiness Institute for Indigenous Health, Dalla Lana School of Public Health, University of Toronto, Toronto, ON M5T 3M7, Canada; 2Nutrition & Dietetics, Dalla Lana School of Public Health, University of Toronto, Toronto, ON M5T 3M7, Canada

**Keywords:** Indigenous, First Nations, housing, environmental or climate health, health promotion, wellbeing, North America

## Abstract

(1) Background: Housing has long been recognized as an essential determinant of health. Our sense of home goes beyond physical shelter and is associated with personal or collective connections with spaces and places. However, modern architecture has gradually lost its connections between people and places; (2) Methods: We examined traditional Indigenous architecture and how it can be utilized in contemporary settings to restore connections to promote the environment, health, and well-being. (3) Results: We found that traditional Indigenous building structures may be the best manifestation of the Indigenous interconnected and holistic worldviews in North America, containing thousands of years of knowledge and wisdom about the land and the connection between humans and the environment, which is the foundation of reciprocal well-being; (4) Conclusions: Learning from the traditional structures, we proposed that modern architects should consider the past, present, and future in every endeavor and design and to utilize traditional knowledge as a crucial source of inspiration in creating works that are beneficial for both current and future generations by taking collectivism, health and well-being, and the environment into consideration in designs.

## 1. Introduction

Housing and the environment have been recognized as important determinants of health [1,2,3,4]. Besides its important physical functions, what makes housing essential to our health and well-being is the connection we make between our ideas of homes, the environment, and collectivism [4,5,6]. ‘Home’ can be considered the place that people live in, but as Blunt and Dowling explain, it is also the feelings imbued in that space that are related to the context of the place [7]. From an Indigenous perspective, home can be the place where wholeness happens. Some people may describe homes as spaces where they can relax, rest, and be themselves, or as places where they can feel a sense of connection, attachment, or belonging, while others may define homes as places to be alongside their loved ones. The common theme is that home is always associated with some personal or collective connections [4]. As described by Blunt and Dowling, our sense of home is shaped by the memories of our childhood, our present experience, and our hopes and expectation of the future [8]. With the home being such a complex, delicate, and intimate concept, a dwelling or building only becomes a home when we make it into an extension and expression of our daily life and identity [4,7]. A good dwelling can help people to feel connected to others and place [6], that they belong (including having an identity within a collective, family, or community) [9,10], and secure. It can foster relationships and support people to live to their full potential. This unique connection between people and place makes a dwelling a crucial factor in determining health and well-being.

More holistically, the dwelling that we, human beings, have shared since time immemorial is Mother Earth. We rely on her to survive and thrive, and we used to give back, as a part of the life cycle, so that balance and harmony were sustained for all living beings [11]. The links between health and the environment are as old as human culture [2,12]. The turning point was the industrial revolution, which enabled significant enhancements in human ‘efficiency’ and ‘productivity’ that changed our relationship with land and the environment [5,7,13,14]. Combined with the propagation of capitalism and consumerism, we changed how we live, work, and play. As significant exploitation of resources and space has been made possible with human development, we have started to tilt the balance. Many Indigenous people believe there needs to be balance in the physical, spiritual, emotional, and mental aspects of self, family, community, nation, and globally, and with all animate and inanimate beings [5,15,16]. The consequences of this tilting balance are the many environmental and health challenges we are facing today. These challenges are also opportunities that ‘force’ us to face the fact that we are a part of a bigger system. Just like Duwamish Chief Seattle said: “Man did not weave the web of life; he is merely a strand in it. Whatever he does to the web, he does to himself.” [12] (p. 15). In other words, the damage we cause to Mother Earth will eventually be seen among human beings. The ultimate determinant of human health is the health of Mother Earth [3,4,12]. Therefore, protecting and nurturing the well-being of the environment is as important as our own well-being. In this case, the question is how can we build our ‘home’ in a way that can best support the well-being of all, including the environment, humans, and all other beings?

In modern, Western society, dominated by scientific rationalism and capitalism, mainstream architecture has gradually lost its connection to place [17,18,19]. When designing residential space, what tends to be first considered is its value as a commodity but not the needs of the future inhabitants and the relationship people can potentially have with the space and place [4,17,20]. There has been an increasing disconnection between architecture and the public [21] which Millette and Oliver [18] say is complex and the root of “colonizing and missionizing” (p. 57). Traditions and cultural contexts have been largely ignored in modern building designs, or only implemented superficially. The intrinsic relationship that people have with their surroundings has been disregarded [21]. This relationship is beyond individuals’ relationship with their living spaces but refers to the reciprocal connection between humans and the natural environment since time immemorial. It comes down to the impacts of human-built structures on the natural surroundings, the living of other beings, and the balance of the ecosystem. Whether we can foster a good relationship between humans and the environment is directly linked to how and where we design and construct our home structures [21].

The connection with the land is rooted in Indigenous Peoples’ cultures and traditions. ‘People of place’ (e.g., people of the Plains, people of the mountains) is how Indigenous Peoples often describe themselves [17,20,22]. With thousands of years living on and with the land, traditional Indigenous architecture designs were based extensively on the traditional knowledge of the land (e.g., the knowledge of the sun, wind, wood, soil, water, and other beings) [19,21]. Traditional lands serve the important physical, social, and spiritual needs of their inhabitants [4]. There were rich considerations behind each construction decision including site selection, building orientation, building materials, and construction techniques. Different traditional dwellings reflect different Indigenous worldviews as well as places that Indigenous Peoples consider traditional territories [4,18,20]. Indigenous traditional architecture is a great manifestation of Indigenous holistic ways of thinking and being. This holistic lens, which is rooted in all relations being interconnected and interdependent, can provide valuable insights on how humans can rebuild the connection between people, environment, and architecture, which in turn promotes the well-being of all: all my relations, all in creation [4].

The literature we found was disparate. Different search strategies were required to understand housing as a determinant of health or environment-centered architectural designs, including North American Indigenous traditional building structures, and even with many search strategies, we found few sources to examine. No studies were found that examined the intersection of all these topics. The objective of this systematic review was (1) to examine North American Indigenous traditional architecture and related traditional knowledge, (2) to understand how, if at all, Indigenous traditional architectural knowledge can be used to inform modern designs and construction, and (3) to understand how, if at all, the incorporation of Indigenous traditional value and knowledge can help to promote the environment and the health and well-being of all populations.

## 2. Materials and Methods

### 2.1. Methodology

The review was conducted following the JBI Systematic Review Guideline for qualitative evidence [6]. This systematic review was executed according to PRISMA guidelines for systematic reviews [7,8]. This systematic review was not preregistered.

### 2.2. Eligibility Criteria

Eligibility criteria were summarized in Table 1. We included articles that met all three criteria: (1) examined indigenous traditional architecture; (2) related to North America Indigenous Peoples; (3) examined the relationship between architecture and health, wellbeing, environment, and Indigenous ways of knowing and being. No criteria were set on the published date, study method, and methodology as they were not applicable to the topic of this review. Journal articles, books, and grey literatures were included.

As one of the inclusion criteria, the term ‘architecture’ was used over ‘building structure’, as ‘architecture’ contains not only building structure, i.e., the products of planning, designing, and constructing of buildings, but also the process, art, philosophy, and the spirit behind physical structures, which were considered, by this research team, as essential in understanding Indigenous traditional structures. We used an Indigenous definition of health: health is physical, mental, emotional, and spiritual, and related to every other aspect of life [8].

### 2.3. Information Sources, Search, Selection

We searched JSTOR, Scopus, PubMed, Medline, Health STAR, Embase, CAB Abstracts, and Avery Index to Architectural Periodicals, Web of Science, and iPortal. Search terms were summarized in Table 2. Both strategies in Table 2 were used to search all databases and Scopus.

Different search terms and strategies were used in iPortal as it did not support advanced search. In iPortal, all peer-reviewed articles under the ‘Dwellings and Architecture’ section under ‘Science and Technology’ were included for screening. Search terms, i.e., ‘Indigenous building’, ‘traditional house’ and ‘architecture’ were used in the iPortal search function, and all results were included for screening.

Except for the search results from iPortal, all search results from databases were exported to Covidence for screening. The search results from iPortal were screened right on the result pages and the selected articles were recorded in Excel sheets for full-text review. All results were screened and full-text reviewed by two authors (R.F. and S.S.) independently. A third author (A.M.P.) helped to validate the selection when there were disagreements. Disagreements were resolved by consensus.

### 2.4. Study Selection

The selection process was summarized in Figure 1. After initial searches, a total of 977 sources were identified. Among these, 872 sources were identified to be irrelevant, and 56 sources proceeded to full-text review. Among the 49 sources retrieved, 45 sources were excluded and four sources, including one book and three journal articles, were selected. Thirty-six book chapters from the book were further screened and reviewed by the two authors (R.F., and S.S.) while three chapters were found to be eligible for the study (See Figure 2).

### 2.5. Data Extraction and Synthesis

One researcher extracted data from each article. Due to the nature of the topic, it was not applicable to conduct critical appraisal or bias and certainty assessments for the included sources. Data extraction was performed manually without using any extraction tools. The researcher extracted data on Indigenous Peoples or regions, traditional building types, study methods, and types of architectural projects. For each included article, the team also extracted the authors’ background information from the internet. This was to respect Indigenous identity and to understand authors’ positionalities in writing about Indigenous topics. All other public building projects, including cultural centers, museums, student centers, and symbol architecture, were excluded in the review, as those structures required different considerations from the structures in which people would live and heal. Thematic analysis was used to analyze data.

### 2.6. Positionality Statement of the Authors

The author would like to make positionality statements, as a proper protocol in Indigenous literature, to allow readers to better understand the perspectives and experiences that informed this review.

Angela Mashford-Pringle is an Algonquin (Timiskaming First Nation) assistant professor and Associate Director at the Waakebiness Institute for Indigenous Health, Dalla Lana School of Public Health (DLSPH) at the University of Toronto.

Ruofan Fu is a settler of Chinese ancestry. She is a registered dietitian and a Master of Public Health student at DLSPH. Ruofan is currently working as a research assistant with Dr. Mashford-Pringle in Indigenous health. Ruofan aims to bring back a holistic understanding of food and food systems in dietetic practice.

Sterling Stutz is a white settler of Ashkenazi Jewish and Irish/English ancestry. She is currently working as the research officer at the AMP Lab in the Waakebiness Institute for Indigenous Health. Sterling’s works are related to cultural safety, health policy, and settlers’ responsibilities in social determinants of health.

As no author has a professional background in architecture design or engineering, this study was approached entirely from a health perspective, aiming to explore Indigenous ways of knowing and being through traditional Indigenous building structures.

## 3. Results

The characteristics of each article and the authors’ background information were summarized in Table 3. Among the six articles, three articles had authors with Indigenous backgrounds [21,23,24]. All authors had backgrounds in architecture and education, except Barbara Oldershaw [25], whose information was not found on the internet. Having Indigenous authors can elevate Indigenous ways of knowing, doing, and being that can authenticate the Indigenous knowledge provided.

Indigenous groups mentioned in each article were found in Canada and the United States, including Navajo, Salish, Nisga’a, Yurok, and Blackfoot Peoples. However, there were two articles that discussed Indigenous architecture in a broad sense instead of focusing on the architectural works of a specific community [21,26]. Indigenous traditional structures discussed in the articles included hooghan, chaha’oh, plank house or longhouse, and tipi.

**Table 3 ijerph-20-04761-t003:** Article Characteristics.

Article	Author Background	Method Indigenous Peoples or Regions Indigenous Traditional Building Structure Key Points
Dalla Costa [27]	Wanda Dalla Costa [27] A member of the Saddle Lake First Nation in Alberta An architect and a professor at Arizona State University Has spent nearly 20 years working with Indigenous communities with a focus on culturally responsive design and built environment as a teaching tool for traditional knowledge	Literature and project review N/A N/A Examined the connection between Indigenous perspectives, knowledge systems, and architecture. Dalla Costa reviewed Indigenous architecture stories from a cultural perspective, discussing four influences in Indigenous architecture (i.e., place, kinship, transformation, and sovereignty), and reviewed three culturally based measurement systems for the design of the built environment. Dalla Costa discussed three contemporary Indigenous architecture examples in Canada. However, these projects were not included in this review as they were not for residence or health purposes.
Glenn [28]	Daniel J. Glenn [28] A member of Apsáalooke (Crow) Nation of Montana The principal architect of 7 Directions Architects, which is a Native American-owned firm based in Seattle, Washington Has more than 30 years of experience in architectural practice. Specializes in culturally and environmentally responsive architecture and planning Has taught architectural design at the University of Washington, Arizona State University, Montana State University and the Boston Architectural Center	Review of projects Navajo and Salish Peoples Hooghan and plank house Glen discussed the fundamental challenge faced by Indigenous tribes across the country: the loss of cultural identity and a struggle to preserve, regain, and continue to evolve Indigenous culture in architecture. Five projects developed by Glen and his team were reviewed. Answering the question: how can we meet the real needs of the present generation of Indigenous Peoples while honoring, respecting, preserving, and celebrating the ancient traditions of ancestors, at the same time, taking the future generation into consideration? Two residential projects, ‘The Nageezi House’ and ‘The Place of Hidden Waters’, were included in this review.
Mackin and Nyce [26]	Nancy Mackin [29] A member of Nisga’a and Tsimshian An architect from British Columbia Taught in Indigenous communities throughout Northwestern Canada, focusing on landscape and building design of Indigenous Peoples Deanna Nyce [30] CEO of the Wilp Wilxo’xskwhl Nisga’a (Nisga’a university-college) Has a background in education, and advocated for Aboriginal education with the British Columbia Ministry of Education A guest lecturer/instructor at many BC education institutions	Literature review and interviews with community members and elders including talking circle Nisga’a community at the northwestern coast of British Columbia, Canada Longhouse and Elder housing A research project supported by Canada Mortgage and Housing Corporation (CMHC). Mackin and Nyce explored the impacts of architecture on health and wellbeing with a focus on built-form solutions for Indigenous Elders living in remote communities.
Malnar and Vodvarka [31]	Joy Monice Marlnar [31] A retired professor. Taught in the faculty school of architecture at the University of Illinois Urbana-Champaign for two decades Specialized in sensory design: how and why we experience spaces and places in the way we do In her later years, turned to studying new architecture on land belonging to Indigenous Peoples in North American Co-author of the book ‘New Architecture on Indigenous Lands’ Frank Vodvarka [31] A professor of fine arts at Loyola University Chicago Taught design, color theory, photography, and history of architecture Co-author of the book ‘New Architecture on Indigenous Lands’	Project review and interviews with architects and habitants Yurok Plank house Malnar and Vodvarka examined four contemporary architecture projects. All four projects were situated on Indigenous lands with local Indigenous authorities involved in every aspect of the creation, from design and construction to financing and programming. Building structure and materials were carefully chosen referring to the traditions with attention to their symbolic meanings. At the same time, contemporary structural capabilities and technology were used to meet the contemporary needs of people. In the analysis, Malnar and Vodvarka not only interviewed the architects behind these projects but also included statements from the inhabitants themselves.
Oldershaw [32]	No information was found.	Literature review Blackfoot Tipi Oldershaw explored the significant roles of tipi in the culture of Blackfeet Peoples. The tipi structure and construction process were explored and analyzed. It was recognized that a tipi is much more than shelter for Blackfeet Peoples. The spatial design enabled a deep connection between its residents and the environment. Tipi also have a great symbolic significance; they are a manifestation of faith and can be a source of emotional comfort and reassurance. Blackfeet women were fully responsible for the construction of tipi. As the builder of the sacred identity, women have high status within the tribe. There was a balance of roles between women and men in Blackfeet culture, where women were the builder of the tribe and men were responsible for supplying the tribe through a buffalo hunt.
Verderber, et al. [33]	Stephen Verderber [34] The Director of the Centre for Design + Health Innovation Registered architect specialized in architecture, design therapeutics, and health A professor at both the John H. Daniels Faculty of Architecture, Landscapes and Design and the Dalla Lana School of Public Health at the University of Toronto	Review of architecture prototypes First Nation Communities northern Canada (i.e., Dene People, Yellowknife, Inuvik, Tuktoyaktuk) N/A Verderber used two architecture case studies, a residential behavioral health residential treatment center and an Elder house design, to demonstrate the application of salutogenesis and eco-humanism in architecture. The two cases were inspired by First Nations’ traditions and developed based on occupants’ needs and ecological considerations. Traditional Indigenous amenities, such as tipi, firepit, healing or sweating lodges were used to complement the design.

### 3.1. Identified Themes

#### 3.1.1. Theme 1: Indigenous Traditional Building Structures

Indigenous traditional building structures discussed in the selected articles included Blackfoot tipi, Nisga’a longhouse [26], Salish plank house [17], and Navajo hooghan and chaha’oh [25]. The features of these traditional building structures along with their designs and construction processes were discussed in selected articles.

A tipi is a traditional structure for the First Nations people of the Plains (see Figure 3). The traditional tipi was cone-shaped, an average of 14 to 16 feet in diameter, and was about 17 feet tall [32]. The framework of the structure was usually built with 20 wooden poles and the tipi cover was usually made of 12 to 14 fabricated buffalo hides which were once plentiful on the Plains [32]. One tipi structure could create a 150–200 square feet space and house a family of eight [32]. In the winter, sod and snow were used to pack the bottom exterior edge of the tipi to prevent excessive draft. In the summer, the bottom of the tipi can be unstacked and rolled up about one foot to increase ventilation [32]. The simple design of tipi made the structure very flexible and reflected an ever-changing nomadic life. Minor refinements have been made in the design of the tipi to protect inhabitants from harsh weather. One example was the dew cloth, which was an interior lining that was hung by ropes tied to the middle of the tipi poles [32]. They were usually exceptionally long and covered down to the bottom of a tipi. The dew cloth can help to prevent rain from dripping off the poles and onto people or items in the interior [32]. The air space between the tipi cover and the dew cloth also created insulation, keeping the tipi warmer in the winter and cooler in the summer. The dew cloth also helped to promote ventilation in the tipi. As the warm air rises in the tipi, it brought in cold air from outside through the space between the dew cloth and the tipi cover. This movement of air created a great path to remove smoke from the central fire [32].

The longhouse (see Figure 4) and plank house (see Figure 5) structures belong to Indigenous Peoples along the shores and rivers all over the coastal regions of the Pacific Ocean on Turtle Island. These houses were used for at least 5000 years, as evidenced archaeologically [25,26]. Both the longhouse and the plank house were pole-and-beam log structures designed to house several generations together under the same roof, so that families could better take care of each other. The built form was designed taking into consideration the needs and activities of both Elders and younger generations. The structure varied significantly in size based on the needs of the communities, from housing four to five to a dozen and more generations [26]. Slightly different from the longhouse, the plank house was a form of modular multi-family housing [25]. New family members can add to the structure by extending another module. Privacy was maintained by hanging woven bark or raising wooden platforms, separating sleeping areas [21]. When each plank unit was put together as a longhouse, it can provide a warm and safe living space for people in the winter [25]. In the summer, plank houses can be removed, put on canoes, and transported to summer camp where they can be transformed into lean-to structures, which was more convenient for summer fishing, berry gathering, and hunting [25].

The traditional dwelling of the Navajo People was the hooghan (see Figure 6), which was a dome-shaped, circular structure with a fire at the center, a central smoke hole, and an east-facing doorway [25]. Navajos’ traditional land was on a high desert mesa 6000 feet above sea level, therefore had strong west wind and extreme temperature change throughout the day [25]. The traditional hooghan was very efficient in handling these extreme conditions. To guard against wind and cold, openings for air and light were limited to the east-facing doorway and a single roof opening [25]. The wall of the structure was made of stone or wood while the roof was made of heavy timber covered in earth [25]. In the summer, another structure, a chaha’oh (see Figure 7), was built to provide a cooler outside space for cooking and sleeping [25]. The chaha’oh structure was built with a four-pole timber frame, which allowed full shade and open sides for the breeze [25].

The Indigenous traditional architecture was one branch of the traditional Indigenous ecological knowledge (TEK) and wisdom [26]. Indigenous Peoples used their architectural knowledge and construction systems, which have undergone thousands of years of testing and experimentation. This knowledge contributed greatly to the maintenance of the diversity and abundance of ecosystems. Materials used in traditional buildings were primarily derived from plant sources and were carefully selected based on the ‘productivity’ of the habitats of the region [26]. Based on Nisga’a collective history, the link between human and the environment were reciprocal. A lack of respect for nature would have devastating consequences on humans’ well-being. Like Elder Hubert McMillan [26] (p. 12) said: “the forests and the rivers set the rules.” Indigenous people took exactly what they needed. Depending on the purpose of use, Indigenous people were always very selective of the type of trees or plants they cut based on their unique characteristics, considering durability, hardness, and usefulness [26]. For example, cedar was considered the prince of trees for the Northwest coastal Nisga’a People [26]. Its lightweight, straight-grained, and rot-resistant long wood made it a perfect material for building longhouses [26]. Its bark (woven into rope), pitch (used as glue), roots (for tying), and wedges were all used in longhouse construction [26]. Indigenous Elders were also very knowledgeable in how different materials could work together as an integrated system to maximize the lifespan of a structure [26].

The methods of construction were established through thousands of years of “learning by doing” [26]. They were based on respect and knowledge of the environment. Knowledge about the construction of traditional structures was conveyed orally and practically from Elders to the younger generation. For example, in mountain regions where the wind was strong, Elders would teach the young how to choose the size and placement of materials, based on the knowledge of sun and wind, to build wind-resistant hunting sheds [26]. This was not simply a construction process but also a spiritual moment for Elders to pass on traditional teaching to the next generation [26].

#### 3.1.2. Theme 2: Beyond Shelters

Traditional structures were designed and constructed in a way that best served all physical, social, cultural, and spiritual needs of Indigenous people [4]. Therefore, they were much more than shelters, but also a reflection of culture and worldview, and revealed each community’s unique ways of thinking and knowledge systems [26,32]. For example, the design of longhouses from the coastal regions reflected the value of family and community [26]. Nisga’a People believed that family together made the community stronger; therefore, the house was designed so that one big family of several generations could live under the same roof comfortably taking care of each other [26]. When families were allowed to live closely together, Elders were able to stay connected with the support and care of their extended families; Elders living with their families tended to have better health and longevity [26]. As the Knowledge Keepers of the community, the health of the Elders was the foundation of the well-being of the community.

Traditional structures also played a vital role in traditional food preparation. Structures were built to accommodate seasonal food gathering and preparation activities [26]. Nisga’a people had special structures for food preparation. Some examples were smokehouses, wind-resistant hunting lean-tos and sheds (wilp-doos), berry-drying racks, and underground storage sheds [26]. All of these structures enabled a year-round abundance of a variety of foods. Teaching was embedded in the process of traditional food preparation. The youth of the community learned the entire cycle of food while having fun and creating core memories with the community [26]. In Nisga’a communities, smokehouses were located close to Elders’ homes, so that they could teach young people the design features of the structure and special strategies to smoke fish [26].

The traditional building itself contained knowledge and stored history. For many Indigenous Peoples, buildings functioned both as utilitarian and sacred spiritual identities whereby ceremonies and knowledge were passed among generations [32]. Looking at Nisga’a culture, carvings and paintings were put on the interior and exterior walls and columns of the longhouses, describing important family events and knowledge [21,26]. In Cree teachings, each tipi pole represented a value, e.g., kinship, kindness, or sharing. The act of erecting a tipi was a way to embrace these values into everyday life and be reminded of the teachings associated with each pole in various aspects of life [26].

Like the Cree, the tipi was also the main dwelling for the Blackfeet people. In the Blackfeet culture, the circle was considered sacred, meaning there is no beginning or end. Blackfeet people believed that “everything the Power of the World does is done in a circle” [32] (p. 44). Like the life of a man, it is a circle from childhood to childhood [32]. Due to this belief being associated with circles, all tipis were considered sacred objects. They represented the ancestors with whom the Blackfeet people maintained close contact [32]. Consequently, tipis were important locations for spiritual ceremonies and prayers [32]. In addition to the spiritual identity, a tipi also provided a unique spatial experience. Like the characteristics of the open Plains, a tipi has no edge to interrupt the flow of space. As was commented by Weatherford (cited by Verderber et al. [33] (p. 44)), it was “like being inside of a geometric molecule and tumbling through the vastness of the Universe.” On the other hand, tipis also facilitated a sense of connection between the dwelling and the surrounding environment [33]. Living in the structure, people could follow the movement of the sun by the light filtered through the cover; the sense of prairie and forest could be captured in the scent of a smudge made of sacred trees and plants (sacred medicines), while the wildlife of the Prairie also existed within the tipi cover [20,33]. As tipis were not soundproof, the noises of the camp, even when one was enclosed in a tipi, could still be heard and the connection with the surrounding community and the land felt.

#### 3.1.3. Theme 3: Colonial History and Indigenous Architecture

One theme that was highlighted in four out of the six articles was the impact of colonialism on Indigenous Peoples’ housing situation. By the late 19th century, many Indigenous people were forced to move away from their homelands and assimilated into Western-style housing through legislation, such as the Indian Act [18,20,35]. For example, in the United States, the Bureau of Indian Affairs believed that buildings like plank houses encouraged communism among Indigenous Peoples and could take away their incentive to work, and “it was not right or best for so great a number of people to live in one great house” [25] (p. 752). Thus, Salish families were forced to abandon their homes and live in “little houses such as white people have” [25] (p. 752). By the end of the 1900s, most plank houses were either abandoned or burned down by the United States Army [25]. Only a few plank houses survive today as ceremonial structures.

As Indigenous people were forced to live within Western structures, significant problems threatened their well-being. For example, the Navajo Peoples’ new ‘homes’ were built based on the Second World War veteran housing plans by the United States federal government [25]. The housing was designed with little or no consultation with Navajo people; therefore, there was no consideration of a multi-generational lifestyle and cultural activities [25]. The compartmentalization of the household does not meet the needs of a more communal way of life. Climate and local resources were also rarely considered in the construction of the newly built ‘homes’ for Indigenous Peoples. For example, the same materials and housing plan were used for both Apache families in southern Arizona and Blackfeet families in northern Montana, resulting in the Apache family sweltering and the Blackfeet family shivering, with expensive heat bills [25]. Historically, the federal housing policy did not respond well to the diverse needs for housing of Indigenous Peoples in Canada and the United States.

### 3.2. Contemporary Indigenous Architecture

As was proposed by Glenn [25], the challenges of contemporary Indigenous architecture are: (1) how can we create residential architecture that reflects unique tribal culture and climate in the contemporary world?; (2) how can we generate architectural works that meet the real needs of the current generation of Indigenous Peoples while respecting and preserving the culture and traditions of the ancestors?; and (3) how can we make architecture design that takes the future generations into consideration? Glenn’s architectural philosophy is rooted in the seven-generation teaching, which taught people to consider the impact of seven generations in every deliberation [25]. The seven generations indicate our understanding of and connection to the three generations of our ancestors, the present generation, and three generations of our descendants [25].

Seven architecture projects from the articles were included in this review and were categorized into four categories based on their designs: (1) projects using a Western architectural approach, (2) projects that combined Western and Indigenous traditional structures, (3) projects based on Indigenous traditional building structures, and (4) projects designed from consultation based on the needs of the future resident.

#### 3.2.1. Project Using a Western Architectural Approach

One architectural solution proposed in Mackin and Nyce’s article was the idea of a cultural village [26]. A modern village was proposed to function like the traditional longhouse, with each unit located close to each other using adaptable housing structures. Individual units were like areas in the longhouse. The flex-housing approach, which had a design feature that emphasized adaptability, could be used for each unit, so that structures could adapt to the different needs of people at different life stages. Using the idea of flex-housing, modern village housing could accommodate the changing lives of individuals and families in a way that was like the traditional longhouse while bringing families and communities closer together [26]. This is one example that used the Western architecture approach to mimic the functions of a traditional building.

#### 3.2.2. Project That Combined Western and Indigenous Traditional Structures

One project led by Glenn was for the Augustine families of the Navajo Nation [25]. It utilized both Western and Indigenous structures to construct homes for the community. As was mentioned previously, the original homes built for Navajo people on a reserve were designed based on veterans’ housing plans built by the federal government in the mid-1960s. According to the 2011 Housing Needs Assessment, more than half of the residential houses in the Navajo Nation required serious repair [26]. The original purpose of the project was to work with Navajo families to repair and renovate their homes; however, considering the conditions of the homes, and after consultation with the Augustine family, it was determined that the best solution was to demolish and rebuild all the homes [25]. The design was developed through ongoing discussion with the Augustine families [25]. The involvement of Navajo students in the project was vital to its success, as they helped the team to gain the trust of the community and Elders while communicating in their Navajo language [25]. The potential to build modern or traditional hooghan and chaha’oh was also explored and discussed. However, except for some Navajo Elders, who grew up in hooghan, most Navajo families had been living in Western-style houses most of their lives [25]. Hooghan had been only used as a ceremonial structure rather than a primary dwelling. Glenn and his team found that most of the Augustine families were not interested in returning to their traditional dwelling; they wanted a more conventional contemporary house but still strongly connected to their traditional culture [25]. The result was a hybrid of the original homes: the hooghan and the chaha’oh. The final design used the original floor plan, which had a conventional division of space, and retained the modern conventions of privacy. However, like a hooghan, the home door faced east, hidden away from the west wind [25]. Each room of the home was wrapped around like an open version of the traditional hooghan [25]. An octagonal courtyard was built with eight juniper logs while a traditional woven corbelled log pattern of the hooghan was used to protect the courtyard from the sun [25]. An outdoor fire was put at the center of each courtyard. Additionally, each building was designed facing south, with a chaha’oh structure providing shade and protecting the windows that face the sun for passive solar gain [25].

To maximize energy use, an energy model was used to optimize material choices, building systems, openings, and orientation. The result was the use of a local aerated concrete material produced by an enterprise of the Navajo Housing Authority, which was predicted to be able to perform the better in the high desert environment than the conventional wood frame [25]. A radiant floor heating system was also implemented to augment the use of solar energy through a passive solar design [25].

#### 3.2.3. Projects Based on Indigenous Traditional Building Structures

Two contemporary architecture projects described were designed based on Indigenous traditional dwelling structures. One of the projects was led by Glenn [25] and his team and designed for the Salish people of the Puget Sound region. Glenn was hired to help to explore how to best utilize a forest site that belonged to the Puyallup Tribe, located in a suburban neighborhood of the city of Tacoma. The site was inaccessible due to the infestation of a non-native blackberry bush and had been used as a dumping ground for old furniture [25]. However, the site retained its natural beauty and was connected to the water of the Puget Sound, which had been the lifeblood of the Puyallup people [25]. Through many discussions with the community members and consideration of a funding opportunity, the team and community eventually decided to make the site into an expanded housing community [25]. Through a series of workshops, a sustainable approach was proposed for the project. The goals were set clearly stating that the design needed to be (1) culturally responsive; (2) be able to foster community; (3) enhance well-being; (4) protect wildlife habitat; (5) reduce energy consumption; and (6) create a safe environment for the people [25]. The building structure needed to use safe materials, be durable with a low-maintenance cost and be able to provide clear indoor air and ample natural light [25].

To achieve a culturally responsive design, the team conducted a thorough study on both traditional and contemporary Puyallup culture and traditional designs [25]. They studied the local climate and analyzed the site from both natural and cultural perspectives. The traditional plank house was designed to allow a secure shared environment for several families living together with privacy maintained by hanging mats separating sleeping areas. However, today’s Puyallup tribal members had developed very distinct ideas of privacy after living in compartmentalized structures for more than a century [25]. Therefore, the traditional plank house structure was not feasible nor desirable for the current generation of Puyallup members [25]. The final design took these factors into consideration and developed a modern version of a traditional plank house. The sleeping areas were designed as fully enclosed housing units laid on either side of the shared spaces [25]. The design was like a courtyard house with several housing units sharing a single linear, semi-covered courtyard [25]. This design also provided a solution to the challenge of maximizing units on the site while limiting the footprint of the new housing to retain as much forest space as possible.

To reduce energy consumption, photovoltaic panels were put on the shed-style roof and window walls for passive solar gain [25]. Cedar was used as the core material for the building, mimicking the traditional plank house. However, despite cedar’s natural durability in the regional climate, the costs of the material and its maintenance were concerning. In this case, a cementitious fiber board with a locally panelized material was chosen [25]. Together, this was an affordable, durable structure with minimal maintenance and better insulation to reduce the energy use of the building. Natural cedar was only used in small areas of the building to maximize the material experience of the resident [25].

The other project was designed by a Western architect, Bob Weisenbach [17]. He worked with Dale Ann Frye Sherman, a cultural consultant, to design the Potawot Health Village [17]. The Health Village came from the idea that the community’s culture came from the rivers and environment and that a healthy environment meant healthy people [17]. The Health Village was an Indigenous version of a healthcare center that was shared by the Yurok, Hoopa, Tolowa, Wiyot, and Karuk tribes. Tribe members were actively engaged in the design process. Instead of having one large impersonal structure like most Western health centers or hospitals, Weisenbach and Sherman decided to create five to six separate structures adapted from the structure of a plank house [17]. Each structure can be a department, and each department was connected by a major circulation route throughout the whole ‘village’. A bonus of such a design was the flexibility to build more individual units as needed. The ‘village’ was designed to form around the stream that flows through it and onwards, resembling the primordial native lands that these once were and mimicking a traditional village [17]. This close relationship between the Health Village and the water was regarded by the tribes as vital in the healing process [17]. Redwood was structurally and culturally symbolic material for the tribes. However, like cedar, it is no longer readily available at a reasonable price. Concrete panels were used for the exterior, while redwood was only used in some essential healing spaces [17].

#### 3.2.4. Projects Designed from Scratch Based on the Needs of the Future Resident

Among the selected articles, three architecture projects were designed from consultation based on the needs of future Indigenous inhabitants and users. Two projects were architecture prototypes discussed by Verderber et al. [33] to demonstrate the application of eco-humanism and salutogenic approach in architecture design. Eco-humanism was the term used by Verderber et al. [33] to describe the Indigenous concept of the interdependency of everything. Verderber et al. [33] argued that eco-humanist designs were meant to support residents’ functional performance and aspiration in both the short- and long-term; at the same time, the designs should be ecologically sustainable, responsive, and adaptable without compromising ecological and social attributes [33], while a salutogenic approach in architecture is centering a design around the health and wellbeing of its residents. The concept of well-being was examined by Verderber et al. [33] on multiple levels, including its physical, social, cultural, and ecological aspects. Together, this meant a more holistic definition of salutogenesis, compared to the other definitions in the literature to date [33].

The first design prototype was a behavior health rehabilitation center located one hour east of Yellowknife in the far north of Canada, designed by Erik Skouris [33]. The aim of the project was to best enable the Dene First Nations’ traditional teaching of healing and ceremonies of knowledge in the treatment of behavioral health or substance abuse-related disorders [33]. Based on the salutogenic concept, a participatory design process was proposed, and a sustainable environmental control system was recommended to minimize waste and energy use. The structure was circle-shaped, inspired by Indigenous Peoples’ unique spatial comprehension of physical space and their deep-rooted connection with the land [33]. The center was radial, which allowed a safe enclosure and established an inward focus that was further reinforced by a central courtyard with a healing lodge at the center [33]. The healing lodge was a traditional sweat lodge. This enabled long-standing First Nations’ healing practices. According to Duran and Duran, as cited by Verderber et al. [33], the sweat lodge was found to be effective in suicide prevention and group-therapy contexts. The courtyard also housed several other traditional amenities, including a teepee tent, an open seating area with a fire pit, and a rock garden for group therapy sessions and informal personal use [33].

The second prototype was a community-based Elder housing design by Jake Pauls Wolf [10]. Three communities in the Beaufort Delta area in the Northwest Territory were selected for this project, including Yellowknife, Inuvik, and Tuktoyaktuk [33]. Among all these communities, Elders had been forced to relocate to large health facilities that were usually far from their villages [33]. Thus, the goal of this project was to enable Elders to remain in their home communities, instead of going to government-sponsored LTC facilities [33].

On-site visits and communication with the tribal members were performed before the design [33]. Based on the fieldwork and the senior population size of each village, resident structures of different sizes were developed [33]. Wolf employed a hybrid strategy, combining off-site prefabrication with on-site conventional construction. These small-scale structures were built with a mix of local and nonlocal materials and methods [33]. They can be great alternatives to those large-scale, impersonal, and government-sponsored institutions. The design also followed the concept of openness and indoor–outdoor connectivity and took ecological impacts into consideration [33]. Short corridors were used, and this allowed an abundance of natural light into the indoor space [33]. Traditional amenities were also used in this prototype, including a fire pit, a garden for socialization, rituals, and traditional practices, a wood deck surrounding each residence, a teepee, and an informal ‘camp’ shed [33]. To further involve nature, walking paths and benches were built to better allow residents to engage with seasonal ponds, streams, and wetlands [33].

The last project to mention was a cultural village project designed by the Ouje-Bougoumou Cree Nation on Lake Opemiska, PQ [26]. It was a community-led project that aimed to create a social living environment that harmonized with the natural environment. The master plan was created based on a vision that was developed through community meetings and was articulated by the Elders [26].

Mackin and Nyce [26] did not describe the detailed design of the whole village in the article; however, they did mention how the Elder housing in the village was located close to the cultural and recreation center. This fostered a space for Elders to share time with the young generation and to pass down traditional knowledge [26]. Based on the needs of the Elders, all Elder housing was designed as single-level buildings so that the Elders could have maximized independence and mobility [26]. A circular element was also incorporated into the building design based on Cree traditions [26].

## 4. Discussion

First, it is important to mention that the authors of half of the articles included in this review are Indigenous. In the past, Indigenous scholars, researchers, and community members were poorly engaged in Indigenous-related architectural projects. Therefore, a core Indigenous perspective was missing in most contemporary architectural designs and discussions about shelter for Indigenous Peoples [36]. However, considering the essential role of housing and home in shaping health and well-being for Indigenous Peoples, Indigenous consultation and engagement should be at the center of the designing and constructing of Indigenous home spaces and environments [22]. With half of the perspectives provided by architects who self-identified as Indigenous, this review can provide accurate reflections of Indigenous Peoples’ worldviews and perspectives.

All traditional structures were extremely adaptive to local climate, e.g., changing seasons and extreme weather conditions, and best supported each unique Indigenous lifestyle. Each structure was designed to best serve its residents while ensuring that there was no exploitation of resources and space. It was ensured that cultural protocols and knowledge were incorporated as part of the design and final structure. Designing and constructing structures like these was only possible when the architects and engineers had a thorough and holistic understanding of the land (e.g., of plants, woods, sun, wind, water, soil, and life cycles) [17,20,33].

Indigenous traditional architecture may be the best manifestation of the Indigenous holistic and interconnected worldviews. These traditional structures are much more than just shelters; they express identities (shown in the variation in designs or paintings on Nisga’a longhouses), nourish a sense of belonging (Elders, adults, youth, and children were part of the group and contributed in various manners as part of the collective or group), fostered relationships, allowed for the care of families, and promoted community building [19,25,26,32]. They provided safe and inspiring cultural spaces that enabled experiences that directly contributed to the physical, emotional, mental, and spiritual well-being of Indigenous Peoples [35]. It is also important to highlight that the significance of traditional Indigenous architecture was not limited to the finished structures. The designing and construction process itself was a highly valuable opportunity for people to make connections with the land and to gain knowledge from the land with everything in it. While the Elders taught the young people to design and construct various structures, traditional teaching was passed on, core memories were created, and bonds were strengthened. The process of erecting the structure was the most important step in building connections with all our relations; humans, plants, animals, trees, birds, fish, water, soil [37].

With an understanding of the cultural and spiritual significance of traditional Indigenous architecture, when looking at colonial history, we can understand how forcibly taking traditional land and the forced assimilation into Western dwelling structures posed destructive impacts on Indigenous cultures and peoples’ well-being. It can be seen from the architecture projects for Navajo and Salish People that the hundreds of years of colonial experience changed the lifestyle of Indigenous Peoples as they had to adapt to the Western colonial system and structures to survive. Thus, moving to the present day, as was proposed by Glenn [25], the challenge of designing contemporary dwellings for Indigenous Peoples is to respect, honor, and preserve the cultural traditions and worldviews, while at the same time ensuring the needs of the current generation of Indigenous Peoples are met.

Looking through the contemporary projects reviewed in this study, it was encouraging to see that Indigenous communities were fully engaged in the designing and constructing of all projects. This is an important act of decolonization and a great step to restore the sovereignty of Indigenous Peoples. The spirit of traditional Indigenous architecture was honored and manifested in all the reviewed architectural works. This spirit was rooted in the respect and preservation of the traditions, the needs and values of its residents, and the consideration of the impacts of human-built forms on the environment and future generations.

Based on the results of this systematic review, we did not identify any studies that examined how traditional Indigenous architectural knowledge on Turtle Island can be utilized in contemporary settings to promote the health and well-being of all populations. However, based on the evidence discussed, traditional Indigenous architecture on Turtle Island can be a great guide for modern architectural works to promote the health and well-being of all in creation, including the environment, humans, and all other beings. The key is to be able to see through and beyond any particular forms, structures, or materials, and to explore deep into the spirit of the traditional ways of living, being and knowing, which are rooted in the Indigenous holistic and interconnected ways of knowing.

For the future practice of architecture, especially for residential designs, in which people live their lives and want a sense of home, we recommend the implementation of the Seven-Generation Teachings: to consider the impacts of the past three generations, the present, and the future three generations. The needs, experiences, and values of future residents need to be explored and honored during the design process by incorporating Indigenous traditional ecological knowledge (TEK) and ways of being and doing that are interconnected with the land and environment. A holistic lens is crucial so that architects can see the multiple layers of needs of people at physical, social, spatial, cultural, emotional, mental, and spiritual levels. This can also be called a salutogenic (health-centered) approach as proposed by Verderber et al. [33].

Indigenous cultures, ceremonies, and traditional ways of being, doing, and knowing are interconnected and interdependent on land which assists with balancing the physical, spiritual, emotional, and mental aspects of self, family, community, nation, and world [5,6]. Architects are encouraged to see the connection between humans and the environment, as a strong connection with nature can contribute greatly to the well-being of people. It is important to explore how the built environment can fit into its surroundings in a way that enables human connections and communication with each other and the environment. To achieve this, all involved in the design and construction of structures should be equipped with adequate knowledge of the local surroundings, which can be achieved through site visits and building relationships with local community members. The result of this process will be a design plan that is beneficial for current and future generations to come.

Finally, Indigenous Peoples have lived on Turtle Island (North America) for thousands of years. They are people of the land and have accumulated a vast knowledge of the land, climate, and the life patterns of all other beings. However, with urbanization, the natural environment has changed dramatically. Some of the traditional building structures may no longer be practical for the contemporary environment; however, what is embedded in the structures is timeless. These structures are not simply forms and materials, but mediums that carry thousands of years of wisdom and knowledge of the land with human and everything else in it. With the many environmental and climate challenges we are facing today, we believe that traditional knowledge and cultures are still crucial sources of inspiration and reference for us to move forward.

### Limitations

For this review, the authors examined selected architecture design projects to understand their impacts on health and wellbeing. All the projects were approached from a health and wellness perspective with a limited focus on the details of architecture science, which is also crucial in deciding the performance of the final structure.

Another limitation was that we did not look into articles or post-occupancy evaluation reports that explored the specific health benefits of the architectural projects discussed in this review. However, we believe that the evaluation of residents’ post-occupancy experience is essential to inform future project development. Future research should be conducted with Indigenous communities in North America to best understand individual and community experiences with contemporary structures and how, if at all, the incorporation of traditional Indigenous structures promoted their health and wellbeing.

## 5. Conclusions

Despite some knowledge that has been shared in the literature about traditional Indigenous structures, more work is needed to further understand how Indigenous traditional structures are connected to Indigenous ways of being, knowing, and doing, and how they can be relevant in contemporary settings. The increase in Indigenous authors in this field may increase Indigenous perspectives, knowledge, and discussion of structures while considering health and well-being. Specific examples of the Elder homes or integration of these structures in a modern village can be viewed as including Indigenous ways of knowing, being, and doing in architecture and community planning. Modern modifications to traditional structures for Indigenous homemaking practice should be led by First Nations, Métis, and Inuit Peoples and communities to ensure the cultural appropriateness of the designs. All of the articles suggest the importance of hearing the Indigenous perspectives on homes and including Indigenous architectural design elements in modern homes and buildings. This review can serve as an important background for building relationships with First Nations, Métis, Inuit, American Indian, and Alaskan Native communities across Turtle Island, to support them with knowledge revitalization around their own traditional structures and to explore how these structures can be utilized in contemporary settings to build appropriate homes or temporary shelters for the communities.

Although none of the projects in this review discussed the potential application of traditional Indigenous architectural knowledge in modern residential designs for general populations, based on the discussions around cultural appropriateness, the salutogenic (health-centered) approach, and eco-humanism, these concepts and knowledge derived from traditional Indigenous approach can provide great insight for modern architecture in dealing with environmental and health challenges. Land, environment, and human beings’ impact on Mother Earth is, and will continue to be, important for the health and well-being of all in creation.

## Figures and Tables

**Figure 1 ijerph-20-04761-f001:**
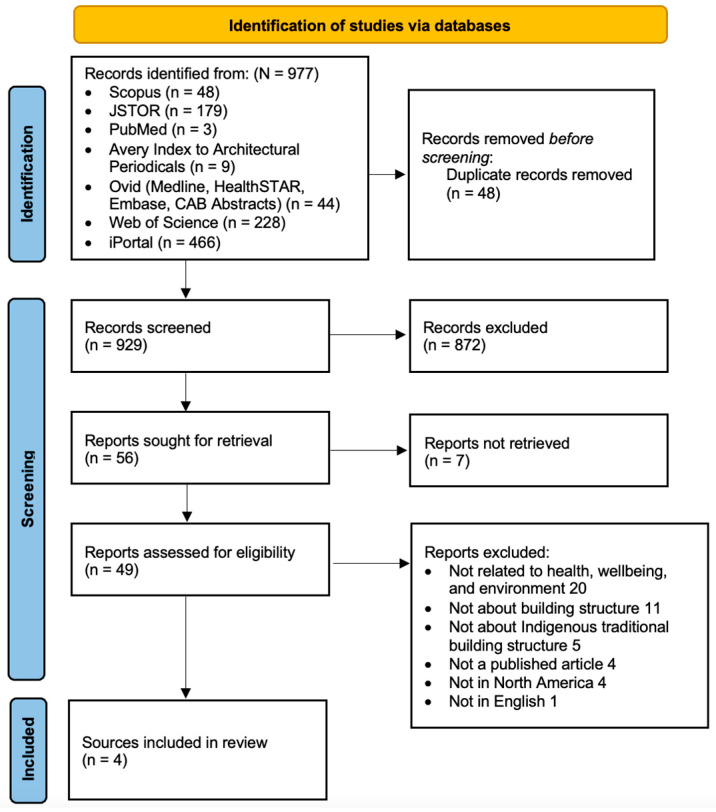
PRISMA flow diagram of the study-selection process.

**Figure 2 ijerph-20-04761-f002:**
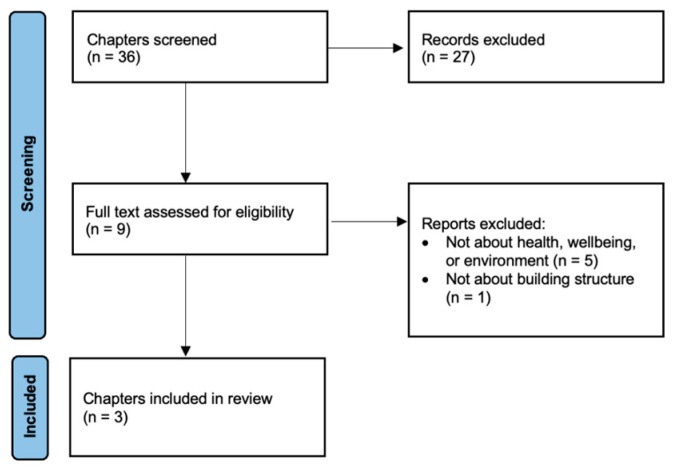
Flow diagram of the book-chapter-selection process.

**Figure 3 ijerph-20-04761-f003:**
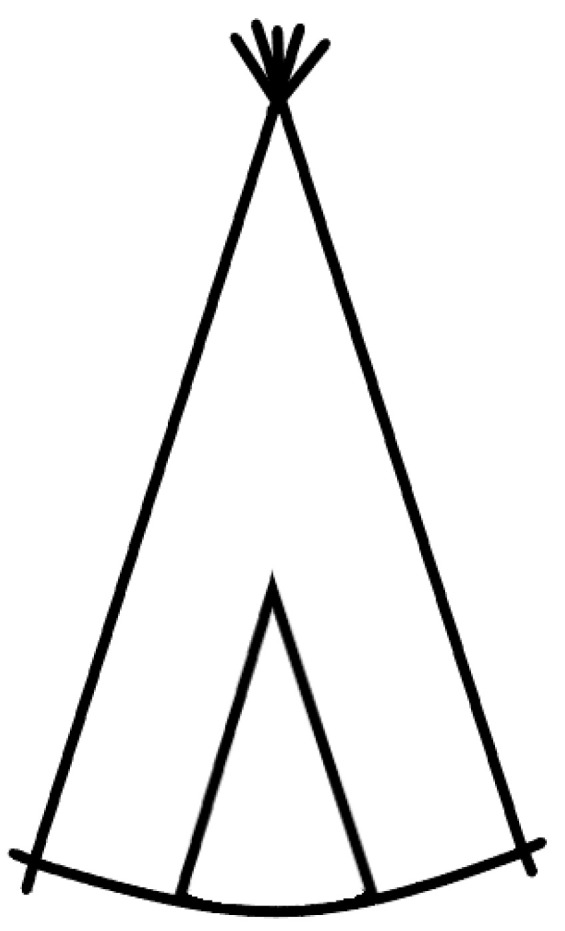
Illustration of a tipi.

**Figure 4 ijerph-20-04761-f004:**
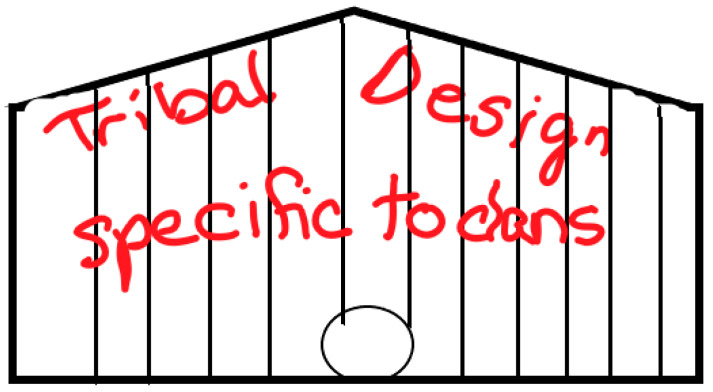
Illustration of Nisga’a longhouse.

**Figure 5 ijerph-20-04761-f005:**
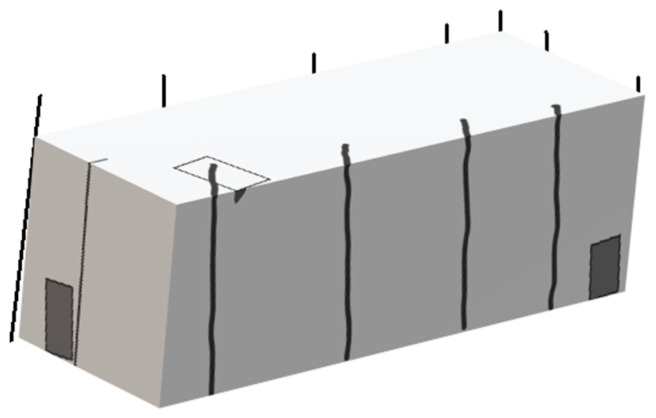
Illustration of Salish plank house.

**Figure 6 ijerph-20-04761-f006:**
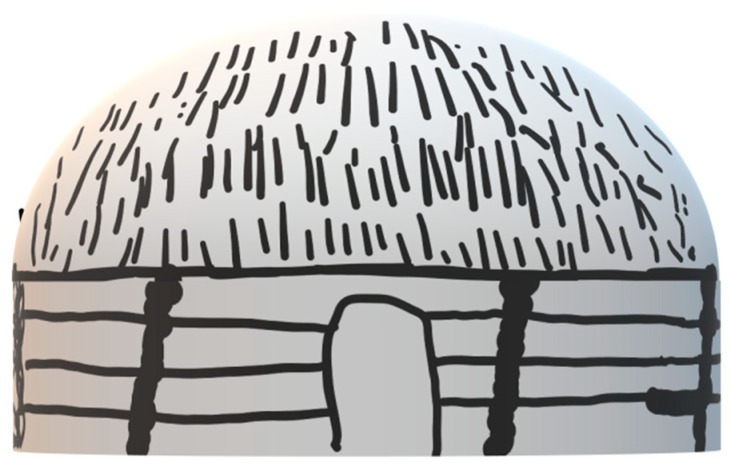
Illustration of Navajo winter hooghan.

**Figure 7 ijerph-20-04761-f007:**
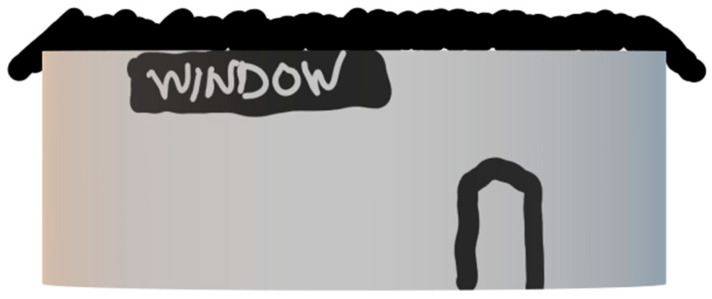
Illustration of Navajo summer chaha’oh.

**Table 1 ijerph-20-04761-t001:** Search Terms.

Context—Culture	Context—Region	Context—Building	Outcome
Indigenous	Canada	hous * design *	health
First Nations	USA	hous * structure *	wellbeing
Native American *	Mexico	building structure *	well-being
American Indian *	America	architecture	environment
		Indigenous hous * design *	
		Indigenous hous * structure *	
		Indigenous building structure *	
		Indigenous building *	
		Indigenous houses	

Note: The * is used as a truncation indicator.

**Table 2 ijerph-20-04761-t002:** Searching strategies June 2022.

Strategy (1)	(Indigenous OR “First Nations” OR “Native American*” OR “American Indian*”) AND (“hous* design*” OR “hous* structure*” OR “building structure*” OR “architecture”) AND (health OR wellbeing OR well-being OR environment) AND (canada OR usa OR mexico OR america)
Strategy (2)	(health OR wellbeing OR well-being OR environment) AND (“indigenous hous* design*” OR “indigenous hous* structure*” OR “indigenous building structure*” OR “indigenous building*” OR “indigenous houses”)

Note: The * is used as a truncation indicator.

## Data Availability

No new data were created or analyzed in this study. Data sharing is not applicable to this article.

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
