# Peer review of "Mamwi Gidaanjitoomin/Together We Build It: A Systematic Review of Traditional Indigenous Building Structures in North America and Their Potential Application in Contemporary Designs to Promote Environment and Well-Being"

_ijerph, 2023, doi:10.3390/ijerph20064761_

Round 1

Reviewer 1 Report

An insightful and well-written paper, which brings unique perspectives for multiple professions related to health and the built environment. The research is original and fills a gap in knowledge about the positive relationships between housing and health in traditional indigenous architecture. Although the authors state that they are not trained in architecture-related fields, they did a wonderful job in understanding and analysing indigenous buildings and their relationships with the environment. 

The following revisions are suggested:

- The overall number of references is very low in the article. Although it is clear that there were limited sources of information for the literature review, additional references could be added to the introduction to support the statements made about broad concepts discussed. In regard to the concepts of holistic wellbeing of indigenous populations, I suggest including papers by Amanda Yates and the 'Mauri Ora compass' she developed - although this was developed for the context of Aotearoa New Zealand, it connects to many general aspects discussed in the paper.

- The formatting of Table 4 could be significantly improved. Some columns could be joined to avoid the large blank space left. It is not easily readable in the current format.

- I would highly recommend including a few images of the buildings discussed in the text, it would make it more interesting and allow a clear visualisation of the concepts discussed. Additional diagrams about the concepts discussed would also enhance the readability of the paper.

- The relationships between architecture and health/wellbeing outcomes were not discussed in depth. Were there any findings about the actual health benefits of the contemporary projects discussed? For example, discussions around health of residents usually include findings from Post-Occupancy Evaluation (POE) surveys. This could be discussed and highlighted as a gap, limitation, and maybe a potential for future research.

- The conclusions could be improved to summarise the main findings and provide more insight into future directions.

Author Response

-The overall number of references is very low in the article. Although it is clear that there were limited sources of information for the literature review, additional references could be added to the introduction to support the statements made about broad concepts discussed. In regard to the concepts of holistic wellbeing of indigenous populations, I suggest including papers by Amanda Yates and the 'Mauri Ora compass' she developed - although this was developed for the context of Aotearoa New Zealand, it connects to many general aspects discussed in the paper.

We have added more citations to the introduction section. Thank you for recommending the paper by Amanda Yates; however, we chose to include only incitation with a North American context.

- The formatting of Table 4 could be significantly improved. Some columns could be joined to avoid the large blank space left. It is not easily readable in the current format.

We have changed Table 4 to be more user-friendly with fewer columns.

- I would highly recommend including a few images of the buildings discussed in the text, it would make it more interesting and allow a clear visualisation of the concepts discussed. Additional diagrams about the concepts discussed would also enhance the readability of the paper.

We have put together some diagrams; however, we would like to discuss with IJERPH how to incorporate the diagrams into the manuscript.

- The relationships between architecture and health/wellbeing outcomes were not discussed in depth. Were there any findings about the actual health benefits of the contemporary projects discussed? For example, discussions around the health of residents usually include findings from Post-Occupancy Evaluation (POE) surveys. This could be discussed and highlighted as a gap, limitation, and maybe a potential for future research.

Unfortunately, this is an area that our team has yet to find. Future research conducted with Indigenous communities in North America would provide some great insights into the benefits of having traditional structures in communities. We have added this to our discussion-limitation section. See lines 1366-1414.

- The conclusions could be improved to summarise the main findings and provide more insight into future directions.

We have improved the conclusion section.

Reviewer 2 Report

This is a significant paper, although it needs to be shortened and better presented, and include illustrations to show the different housing typologies. While methodology and data collection often foregrounds  findings. I think in this instance it would be better to get straight into  the different types of housing  without wading through such a long and slightly pedantic materials methods, search terms and eligibility criteria and selection section (lines 102 to 225). I found this part tedious to read. I kept skipping over it. To make your paper more compelling, start with the themes integrating any relevant author info extracted from table 4 into the text, then concisely explain  how you collected data  by removing or placing other extraneous information, that the reader doesn't need to know in detail (including figures 1, 2 and table 1) from lines 102 to 225 into footnotes or appendix instead.  Then end with  the discussion section. 

Minor edits are required to tighten the readability and remove repeated phrases like "On the intangible level",

Author Response

This is a significant paper, although it needs to be shortened and better presented, and include illustrations to show the different housing typologies. While methodology and data collection often foregrounds findings. I think in this instance it would be better to get straight into the different types of housing without wading through such a long and slightly pedantic materials methods, search terms and eligibility criteria and selection section (lines 102 to 225). I found this part tedious to read. I kept skipping over it. To make your paper more compelling, start with the themes integrating any relevant author info extracted from table 4 into the text, then concisely explain how you collected data by removing or placing other extraneous information, that the reader doesn't need to know in detail (including figures 1, 2 and table 1) from lines 102 to 225 into footnotes or appendix instead.  Then end with the discussion section. 

We have put together some diagrams for traditional building structures. We have shortened the methods section and condensed Table 4.

Minor edits are required to tighten the readability and remove repeated phrases like "On the intangible level",

We have made some minor edits to improve readability.

Reviewer 3 Report

The aim has been rather too general and as a result the outcome remains unclear. The introductory sections 1 and 2 must be rethought. The paper presents some very interesting examples in the Results section (section 3), but there needs to be a comparative method by which to extract some findings that would then suggest the paper’s theme and structure. Also, the list of references is very limited for this project.

Following the above, I wish to make the following remarks hoping to help the authors in setting the paper’s focus:

Rather than attempting to set a clear target, the paper adopts an ideological stance based on an idealistic view about an early time and past condition, which however may not be true. For example, it suggests that at some point we (humans) started to exploit resources and space and that this has caused imbalance. That view is quite problematic. It assumes that at some point we (humans) started to exploit resources, whereas in fact historically that has been the case in most periods. The type and extent of exploitation may have changed, but there is no evidence that humans at some point in history have changed their view or attitude. Similarly, humans may have been acting as “capitalists” much before that term was introduced and likewise, efficiency and productivity has been their goal too, not being exclusive to industrial revolution and the ensuing periods. Additionally, utterances such as “In modern western society, dominated by scientific rationalism and capitalism, capitalism, main stream architecture has gradually lost its connection to place” (line 64) or “there has been an increasing disconnection between architecture and the public” and “the intrinsic relationship that people have with their surroundings has been disregarded” that immediately follow are somewhat too plain, and as a result they lack scientific rigor, not helping the points being raised.

Later in the paper it says that “Traditional lands serve the important physical, social, and spiritual needs of their inhabitants. There were rich considerations behind each construction decision including site selection, building orientation, building materials, and construction techniques” (lines 83-85). This needs to be shown in detail later in the paper. Section 3 and the whole paper need to explore such issues in further depth and similarly, the findings need to be supported by examples, references and visual material. The authors may keep in mind that their knowledge and expertise on indigenous culture is not common to most readers and so they need to be introduced in a more tangible form so that such concepts become the paper’s main theme.

Moreover, statements such as this one: “Different search strategies were required to understand housing as a determinant of health, health or environment-centered architectural designs, and Indigenous traditional building structures. No studies were found that examined the intersection of these topics” (lines 93-96), are not true. Even though such statements offer a humanistic, social, and nature-oriented perspective, they read as being too general, they are not proven and in fact they may not always be true. It is better to ground such statements (and other ones that follow) on facts, rather than on ideological grounds. In fact, under different traditional contexts, such issues have been raised numerous times. A more moderate expressive style and a clearer framing from the start, noting for example how in the North American context such issues have been disregarded, would help to justify the paper’s viewpoints.

The Results section should be expanded and the material they present could be compared and help draw findings and ideas that could be applied in the western North American context. And, as hinted above, the themes and examples being discussed must be supported by photographs and diagrams explaining the claims.

Finally, the parts “Information sources, search, selection,” “Study Selection,” “Data extraction and synthesis” and “Positionality Statement of the Authors” in Section 2 are unnecessarily detailed and read as being redundant.

Author Response

The aim has been rather too general and as a result the outcome remains unclear. The introductory sections 1 and 2 must be rethought. The paper presents some very interesting examples in the Results section (section 3), but there needs to be a comparative method by which to extract some findings that would then suggest the paper’s theme and structure. Also, the list of references is very limited for this project.

The intention of the review and paper was to examine the intersection of health and well-being and Indigenous traditional structures. However, very few papers and books discuss Indigenous architecture beyond planning, design, and useability. Regrettably, the articles and book we found did not specifically look at health, yet we know health is a key determinant in the health and well-being of human beings. We have made changes to highlight this. We have added more references to back up our points.

Following the above, I wish to make the following remarks hoping to help the authors in setting the paper’s focus:

Rather than attempting to set a clear target, the paper adopts an ideological stance based on an idealistic view about an early time and past condition, which however may not be true. For example, it suggests that at some point we (humans) started to exploit resources and space and that this has caused imbalance. That view is quite problematic. It assumes that at some point we (humans) started to exploit resources, whereas in fact historically that has been the case in most periods. The type and extent of exploitation may have changed, but there is no evidence that humans at some point in history have changed their view or attitude. Similarly, humans may have been acting as “capitalists” much before that term was introduced and likewise, efficiency and productivity has been their goal too, not being exclusive to industrial revolution and the ensuing periods. Additionally, utterances such as “In modern western society, dominated by scientific rationalism and capitalism, capitalism, main stream architecture has gradually lost its connection to place” (line 64) or “there has been an increasing disconnection between architecture and the public” and “the intrinsic relationship that people have with their surroundings has been disregarded” that immediately follow are somewhat too plain, and as a result they lack scientific rigor, not helping the points being raised.

We have adjusted our introduction: we made it clear that the industrial revolution was what made it possible for humans to make significant changes to our environment and propagated and amplified the impact of capitalism.

We argue that there was a point during the Industrial Revolution that accelerated the exploitation of resources in North America and caused mental, spiritual, physical, emotional, and social imbalances. This imbalance can be directly associated with the shelter that people inhabited, which did not allow for multi-generational homes and connections to the land that provided food, water, and social connection.

See Lines 162-201 for adjustment.

Later in the paper it says that “Traditional lands serve the important physical, social, and spiritual needs of their inhabitants. There were rich considerations behind each construction decision including site selection, building orientation, building materials, and construction techniques” (lines 83-85). This needs to be shown in detail later in the paper. Section 3 and the whole paper need to explore such issues in further depth and similarly, the findings need to be supported by examples, references and visual material. The authors may keep in mind that their knowledge and expertise on indigenous culture is not common to most readers and so they need to be introduced in a more tangible form so that such concepts become the paper’s main theme.

Lines 83-85 are now lines 229-232. The whole result section of our paper further elaborated on this statement. Based on each traditional structure, we talked about the importance of structures on Indigenous People’s physical (e.g., lines 580-592, 617-618, 623-627, 636, 733-735), social (e.g., lines 728-732), and spiritual health (e.g., lines 721-722, 747-749), and the rich considerations behind site selection (e.g., lines 719), building orientation (e.g., line 673), building materials (e.g., lines 625-654, 657-666), and construction techniques (e.g., lines 664-665).

Moreover, statements such as this one: “Different search strategies were required to understand housing as a determinant of health, health or environment-centered architectural designs, and Indigenous traditional building structures. No studies were found that examined the intersection of these topics” (lines 93-96), are not true. Even though such statements offer a humanistic, social, and nature-oriented perspective, they read as being too general, they are not proven and in fact they may not always be true. It is better to ground such statements (and other ones that follow) on facts, rather than on ideological grounds. In fact, under different traditional contexts, such issues have been raised numerous times. A more moderate expressive style and a clearer framing from the start, noting for example how in the North American context such issues have been disregarded, would help to justify the paper’s viewpoints.

Through our search strategy we did not find any articles that were at the intersection of Indigenous housing/structures, health and environment-centred architectural design. We did not make changes to this section as there was nothing found. It was unclear what we can do to change Lines 93-96.

The Results section should be expanded and the material they present could be compared and help draw findings and ideas that could be applied in the western North American context. And, as hinted above, the themes and examples being discussed must be supported by photographs and diagrams explaining the claims.

We did not expand on the result section, how can we explain here? Comparison of different structures is not what we intend to do in the review. We were trying to identify the essence of the Indigenous way of knowing and being by examining the traditional structures and identifying common themes. We have created diagrams to explain the traditional structures further.

Finally, the parts “Information sources, search, selection,” “Study Selection,” “Data extraction and synthesis” and “Positionality Statement of the Authors” in Section 2 are unnecessarily detailed and read as being redundant.

We have cut down all the parts listed in the method section except for the positionality statement. We believe that positionality statements are very important in Indigenous-related studies so that readers can understand the perspective and experience that informed the review. This is the same reason that we explored into authors’ backgrounds for each article we selected and reviewed for the study.

Round 2

Reviewer 3 Report

The paper has significantly improved since its previous version, so I believe it has reached a level it can be published as is.